# Application of the EFQM Model to Assess the Readiness and Sustainability of the Implementation of I4.0 in Slovakian Companies

**Renata Turisova, Juraj Sinay, Hana Pacaiova \*, Zuzana Kotianova and Juraj Glatz**

Faculty of Mechanical Engineering, Technical University of Kosice, 04200 Kosice, Slovakia;
renata.turisova@tuke.sk (R.T.); juraj.sinay@tuke.sk (J.S.); zuzana.kotianova@tuke.sk (Z.K.);
juraj.glatz@tuke.sk (J.G.)

**\*** Correspondence: hana.pacaiova@tuke.sk; Tel.: +421-903-719-474

**Abstract:** The fourth industrial revolution (I4.0) is expected to increase quality, efficiency, availability, sustainability, the reduction of costs, the demand for energy and environment, and mainly increase the level of occupational health and safety (OHS). New procedures or paradigms of this revolution deflect from already used standards and create an assumption for building the exceptionality of organizations. The main idea of the performed research was to assess how managers in the Slovak industry perceive the readiness of organizations for the implementation of I4.0. The aim of this study, applied in 53 companies, was to assess two areas: the integration level of complex safety into management systems; and the impact of digitalization on OHS. The applied methodology was based on a modified EFQM (European Foundation for Quality Management) exceptionality model. Answers were transformed into numeric figures using a so-called spider web diagram. In the conclusion of this article, there are described interesting differences in the two mentioned areas based on the perception of both top management and the estimation of the readiness degree of the Slovak organizations for I4.0 concept.

**Keywords:** safety; digitalization; integration

---

## 1. Introduction

Most developed industrial countries have, in the last decade, been intensively dealing with the arrival of the so called fourth industrial revolution, Industry 4.0 (I4.0). It is a concept based on such elements as the industrial internet of things, cyber-physical systems, artificial intelligence etc. It seems that the timely capture of the onset of the mentioned industrial revolution is, for particular industrially oriented countries, of essential significance, not only from the point of view of their competitiveness [1,2]. Globalization and risk factors arising therefrom (data safety, information sensitivity and vulnerability, readiness for crisis situations) will verify the effect of digitalization on the continuity of business management [3–10].

Furthermore, in Slovakia, which is industrially oriented, this issue is dealt with by numerous experts from practice, academicians, and often, unfortunately, by politicians too. They come with various opinions, views, knowledge, and determination to implement Industry 4.0 into various areas of Slovak industry. However, there is one fundamental problem—whether Slovak industrial plants are sufficiently ready for the implementation of Industry 4.0. According to the study, from Grenčíková et al. [11] within 80 industries operating in the mechanical engineering sector, in 2017, only 66% of respondents stated that Industry 4.0 is very important for the future, but in 2018 only 59% of respondents had the same opinion. What was interesting, however, was that compared to 2017, 2018 increased the number of companies stated to deal with I4.0 by 14%.

The concept of the fourth industrial revolution is based on the connection of the virtual cybernetical world with the real world, where not only physical laws apply, but also those social, cultural, economic, and other laws [2,12]. This brings about the necessity to identify, recognize, and understand significant interactions between particular systems and the entire society. Thus, a complex cybernetic-physical-social system is created, which is the base of I4.0 [13]. The vision of I4.0 concept functioning expects a deep, knowledge-based industrial integration, applying information and cybernetic technologies. It must be able to massively share a lot of information (big data) and generally, in the real time, continuously communicate with autonomous robots, sensors, cloud, and data storages. Above it all—as an idea and technology core, it must stand the latest and suitable applied methods and processes of cybernetics and artificial intelligence [14,15]. This is why I4.0 is sometimes spoken about as a revolution of "creative thinking" [16,17].

A whole array of scientific and expert articles deal with the issue of Industry 4.0 implementation and its integration of existing and new technologies. Authors Bangemann et al. [15] analyze how to reach an accord between existing and new technologies and machinery equipment using so-called mixed systems. Authors Müller and Voigt [18] in Nurnberg, on a sample of 177 small and medium companies, looked into respondents' possibilities and readiness for the implementation of Industry 4.0 principles, from the point of view of activities, corporate relations, supplier—customer relations and their potential information interconnection. The issue of Industry 4.0 implementation was also dealt with by Sabine Pfeiffer [19]. A study by authors Veile and Kiel [20] is based on empirical data gained from 13 half-structured detailed interviews with German experts, who have experience with the implementation of Industry 4.0. It is one of the first documents to mention concrete examples of observations gained directly from the industrial application of Industry 4.0. Among other issues, it handles the problem of financial fund provision, integration of employees into the integration process, and the creation of open flexible corporate culture. It also handles the question of planning process complexity, cooperation with external partners, correct data interface handling, interdisciplinary communication, organizational structure changes, and data safety.

The mentioned studies and published survey results from 2017 [11] inspired the researchers to assess the degree of readiness of Slovak companies for I4.0 with regard to two areas: integration of complex safety (Safety and Security) into management systems (ISMS) and the impact of digitalization on occupational health and safety (OHSd). For the purposes of the second area of research, under the term of digitalization, all elements which present I4.0 were analyzed, e.g., advanced robotics, additive manufacturing, industrial internet (industrial internet of things IIoT), and clouds and their impact on humans. The questioning scheme stemmed from a partly modified model aimed at the company efficiency measurement EFQM [21–28], as the objective of the study was to verify the assumptions of Slovak companies for Industry 4.0 implementation. From the EFQM model in question, only its Enablers part (without the Results part) was used. The survey was conducted in 53 organizations, out of which 36 organizations focused on automotive industry, or automotive industry suppliers. A detailed description of questioning and the way of evaluation highly exceed the possible length of this contribution, which is why we will further mention only some results gained from the survey in question.

## 2. Materials and Methods

There are many management tools and techniques which can be used as support for the management of an organization in various areas (e.g., project, strategic, knowledge, risk, quality, and safety management) [29]. The exceptionality model EFQM is focused on performance and thus enables a general view on the organization. It also provides a manual on how to use the selected tools, processes, or principles (in our case, these are principles and processes of Industry 4.0 aimed at ISMS and OHSd) so that they synergistically complement each other in order to ensure sustainable development of the organization and increase its profit. This model represents a complex management framework which is used by more than three thousand organizations all over Europe [21]. The model

is regularly revised and updated in three-year cycles based on learning and gaining experience of leading European organizations. As a base of our questioning, we used a so called "classical" model used until 2018 (EFQM, 2018) [21,22]. It is designed in a way so as to be a practical and factual tool enabling organizations to gain an overall review of their current exceptionality level. It should also help determine priorities of their efforts for improvement so that it has a maximum impact. The model is universal, which results in the fact that it can be applied for any organization regardless of its size or specialization [27,28]. In order for an organization to be successful, it needs a strong management and strategic plan. It needs to develop and improve the abilities of its employees, partnerships, and processes in order to gain added value of its products for its customers. As long as the approaches are correctly implemented, the results expected by the parties involved are achieved. The organization needs to realize these given assumptions so that it can implement and develop its strategies on its way to exceptionality.

## 2.1. Exceptionality Criteria

The EFQM exceptionality model is a generalized framework based on nine criteria, by means of which it is possible to perform a thorough evaluation of the exceptionality degree of any organization. The criteria are divided into two areas. The first five criteria are called assumptions. They describe and evaluate what the organization does and how it does it. The remaining four criteria evaluate the achieved results [21,24].

The assumption part of the model was applied to create 5 basic assumption criteria: (1) Leadership; (2) People; (3) Strategy; (4) Partnership and Resources; (5) Processes, Products and Services essential to achieve the exceptionality itself, in our case focused on two areas (ISMS and OHSd) and their relations with Industry 4.0—see Table 1.

During assumption analysis, it is evaluated how the approaches adopted by the organization are used. The approaches should be more or less used and mainly integrated. The model in question also evaluates how these approaches are applied and assesses how these are applicable in the organization as well as their systematic application in all relevant areas. It is also a subject of research by means of self-evaluation if these assumptions are systematically evaluated and improved. Corresponding measurements should be performed, and particular activities should be implemented based on the principles of learning and innovations [30,31].

**Table 1.** Basic assumption criteria of the EFQM model.

| Assumption Criterion | Description |
|---|---|
| Leadership (L) | Assesses:<br>- if these leaders encourage other employees to reach the set goals;<br>- how the management of the organization build culture based on social responsibility stemming from the mission and vision of the organization;<br>- if there are leaders within the organization who are examples of integrity and ethical behavior;<br>- how management members develop the organization management system towards reaching perfect results. |
| People (Pe) | Deals with how:<br>- the organization creates and modifies personnel plans;<br>- recruitment, education and employees' development are run;<br>- the organization motivates employees by the system of remuneration and rewarding;<br>- the organization encourages employees to join the process of organization efficiency improvement;<br>- the transfer of responsibility and authority functions is done;<br>- it assesses suitable working conditions;<br>- carries out employee satisfaction surveys. |

**Table 1.** *Cont.*

| Assumption Criterion | Description |
|---|---|
| Strategy (S) | Assesses:<br>- needs and expectations of involved parties;<br>- how it uses inputs into strategy creation;<br>- how the organization monitors external environment and internal efficiency;<br>- if it performs comparison with suitable benchmarks. |
| Partnership and Resources (Pa) | Deals with how:<br>- the organization establishes, builds and maintains relations with suppliers, customers and other partners;<br>- the organization manages its financial sources, plans investments and maintains its assets;<br>- the organization is able to manage the information and knowledge within the organization;<br>- it guarantees them in the context of intellectual property protection. |
| Processes, Products and Services (Pr) | Deals with:<br>- creation, implementation and subsequent management of processes;<br>- how the organization looks into and evaluates the efficiency of its processes;<br>- how their improvement is running;<br>- how the organization manages the life cycle of its products, from their development through promotion until their delivery to customers;<br>- how customers' feedback is used as an input for product portfolio management. |

## 2.2. Model of Organization Readiness for Industry 4.0

The questionnaire which was used could not be by far as complex as EFQM, which stemmed from organization and self-evaluation, but it was subsequently modified by an expert external evaluator. Our questioning was based only on subjective estimates of the questioned organization's top managers. The range of this questioning did not allow us to use a wide spectrum of questions as as used by self-evaluation using the EFQM exceptionality model. On the other hand, it was not a complex organization evaluation, only the evaluation of exceptionality assumption fulfilled a degree the perception of questioned organizations in the area of integrated safety and digitalization from the top managers' point of view. Therefore, not only did we reduce the number of questions, but we also modified the way of evaluation. From the EFQM model, within the questioning, we adopted the structure of its assumption as well as the spider web diagram [24] at the quantification of particular answers.

The basic EFQM conception is reduced in such a way that it reflects the readiness degree of questioned organizations in two selected areas from the point of view of Industry 4.0 methodology. Specifically, the first part of this questionnaire is focused on respondents' perception from the view of the degree and the way of ISMS and the second part of the questionnaire is focused on OHSd in terms of Industry 4.0 principles (see Appendix A). Both parts can reach the evaluation of between 0 and 50 points. Each question accounts for between 0 and 4. It makes 100% together.

When starting from the assumption that digitalization (OHSd) is an equally essential assumption to ISMS, it was then possible to evaluate how each respondent perceived the mentioned aspects in their organization by a sole evaluation (see Table 2).

Total point evaluations are balanced in terms of EFQM model recommendations in such a way that this total evaluation ranging from 0 to 100 points characterizes such respondent's perception of an organization which corresponds to the latest requirements for safety system management. These, by their integration into the complex management and by using modern tools provided by current information technology, create all assumptions for a reliable and effective efficiency of the systems in question. During the survey, we addressed organizations with a focus on their management. It is the orientation on top managers that, in a certain way, excuses the relatively small respondent sample (53 questionnaires). As we hinted in the previous part, particular respondents' answers

(top managers) were transformed into point evaluation using the range from spider web diagram. In the particular graphic presentation questioning the results, we used percentual result evaluation, where 100% represented the maximum number of points which were possible to reach within the particular criterion. It is necessary to bring to attention that in the part of questioning which used the structure of the EFQM model assumption part, all questions were closed and particular answers were expressly assigned concrete numeral values.

**Table 2.** Point evaluation of ISMS area and OHSd.

| Criterion | ISMS | | | OHSd | | | Total |
|---|---|---|---|---|---|---|---|
| | Num. of Questions | Max. Points | Total Points | Num. of Questions | Max. Points | Total Points | |
| L | 2 | 4 | 8 | 3 | 4 | 12 | 20 |
| Pe | 2 | 4 | 8 | 2 | 4 | 8 | 16 |
| S | 2 | 4 | 11 | 1 | 4 | 7 | 18 |
| | 1 | 3 | | 1 | 3 | | |
| Pa | 1 | 4 | 7 | 2 | 4 | 11 | 18 |
| | 1 | 3 | | 1 | 3 | | |
| Pr | 4 | 4 | 16 | 3 | 4 | 12 | 28 |
| Total | 13 | - | 50 | 13 | - | 50 | 100 |

Note: L—Leadership; Pe—People; S—Strategy; Pa—Partnership and Resources; Pr—Processes, Products and Services; ISMS—Integration of complex Safety (Safety and Security) into Management Systems (hereinafter only as Integrated safety); OHSd—impact of digitalization on OHS (hereinafter only as Digitalization).

## 3. Results

For both areas (Integrated safety = ISMS and Digitalization = OHSd) a relatively high answer variability from 53 respondents is characteristic (see Figure 1 and Appendix A). Whereas, the quartile range at Digitalization (OHSd) represents approximately 40% (between 20 and 60), the organizations reported by their own top managers within the questioning model, were perceived as organizations with almost a zero digitalization degree. However, there were also such ones that evaluated digitalization on the level of 100%. Evaluation median approximately corresponded to the average on the level of 40%. By the ISMS area assessment, the variability of answers was slightly smaller. With approximately the same quartile range, smaller variability was reported and there were no reports of such extreme evaluations as in the case of OHSd. While the evaluation median moved to the level of approximately 60%, the mean value increased to only approximately 50%. In any case, from the respondents' answers, it is possible to observe significantly better evaluation of Integrated safety than in the area of Digitalization.

During the evaluation of the answers to the questioning, we will use approximate (rounded) values of reached per cents. As it is respondents' subjective evaluation, the exact determination of answers with decimal point exactness is not of high importance for result interpretation.

A relatively high variability of answers may be interpreted as a different degree of readiness of questioned organizations in the evaluated areas. A certain distortion of gained numeral figures may be caused by respondents' different ideas about the meaning and degree of digitalization within safety from the Industry 4.0 point of view.

The next figure (Figure 2) shows Boxplot diagrams stratified according to particular criteria in terms of quantitative evaluation of the questioning model structure. It is visible from the criteria evaluation in question that in all assumption criteria, the perception of Integrated safety assumptions is perceptually evaluated on a higher level than the perception of Digitalization assumptions. The variability of answers is equally high in all criteria. It is possible to observe some changes in quartile ranges, mean values, and medians. The assumption focused on "Leadership (L)" criterion is perceived best in both assessed areas. Between them, there is also the smallest movement of medians and mean values. In the Digitalization area, all other criteria (except for Processes) are evaluated with a median on the level of approximately 25%. As for the "Processes, products, and services (Pr)" criterion, it is

approximately 35%. The criteria generally evaluated worst are "Partnership and Resources (Pa)". Within criteria "Employees (Pe)" and "Strategy (S)", there are the biggest differences in evaluation.

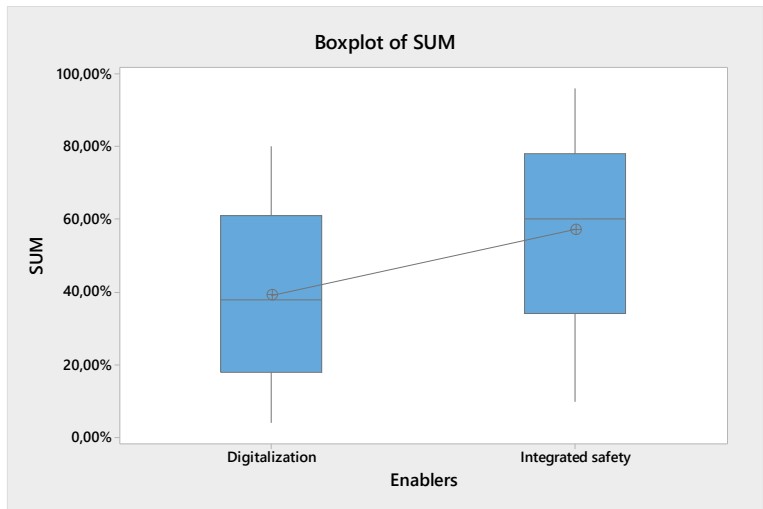

**Figure 1.** Organizations' answers in the area of Integrated safety and Digitalization. (Note: SUM explain the total number of points reached in researched areas figured in percentage). Source: own research.

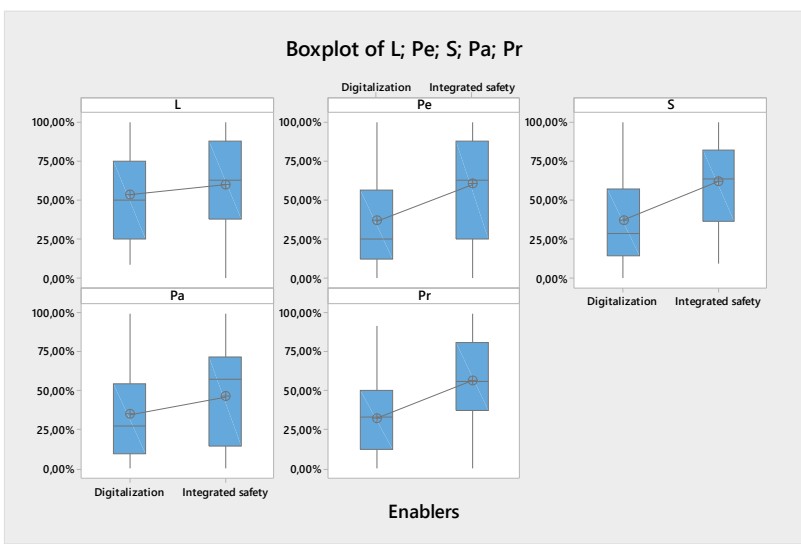

**Figure 2.** Organizations' answers in the area of Integrated safety and Digitalization according to particular criteria. Source: own research.

The best evaluated criterion "Leadership (L)" is possible to be interpreted as a perception of great importance during the Industry 4.0 concept implementation in both researched areas. Especially in the early phases of Industry 4.0 implementation, the importance of leadership is significant. Even though this result may be distorted by the fact that the respondents were top managers, it is generally possible to state that the idea of Industry 4.0, also in the assessed areas, is within the assessed group of organizations communicated by their leaders, and thus positively perceived by respondents. Low evaluation of the "Processes (Pr)" and "Partnerships (Pa)" criteria could possibly be interpreted as a relatively low degree of Industry 4.0 implementation in the researched areas. Big movements of the mean values within the "Employees (Pe)" and "Strategy (S)" criteria may be interpreted as a real difference between the areas of Digitalization and Integrated safety, as for the readiness of employees, education etc., but also in strategic planning. The high variability only illustrates the significant heterogeneity of the perception of particular respondents from various questioned organizations.

Given the mentioned high variability, we added, during the subsequent analysis, another stratification factor—sector. The following four diagrams (see Figures 3–6) show the results of questioning, in the criteria structure according to the questioning model, as well as their division according to sectors, in which the questioned organization works. The relatively highest average respondents' percentual evaluation reached the organizations from the automotive, electrical, and mechanical sector (see Figure 3). In all three mentioned sectors it is possible to observe a high variability of answers reaching, not rarely, the whole range. With the automotive and electrical sectors, the worst evaluated was the "Partnerships and Resources (Pa)" criterion. On the contrary, the "Processes, products, and services (Pr)" criterion, which was generally evaluated on a low level, was with the two mentioned sectors evaluated, on average, relatively highly. With the "Mechanical" sector, the average evaluation of the given criterion is the lowest; however, the second lowest evaluated position is paradoxically the "Leadership (L)" criterion.

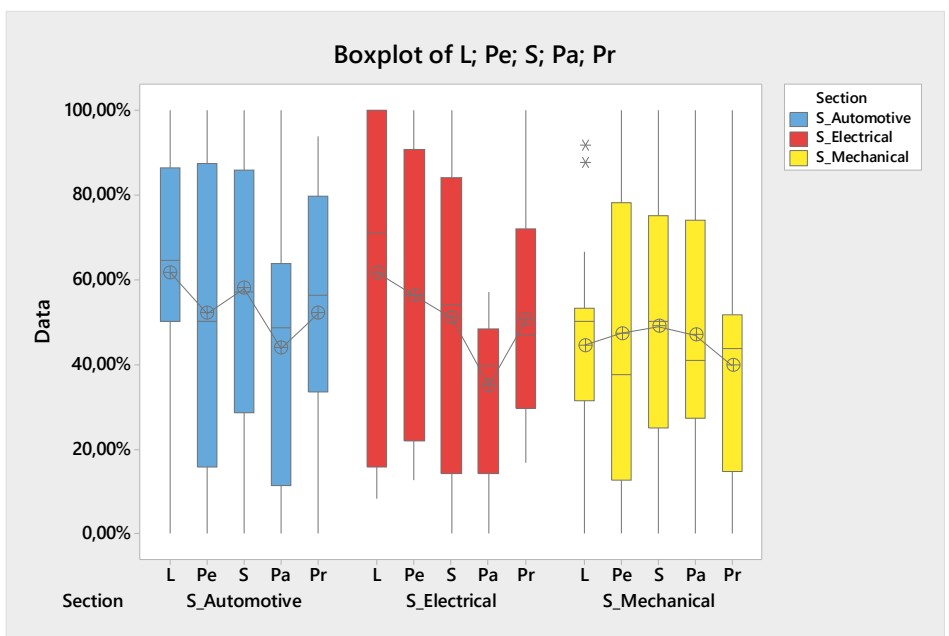

**Figure 3.** Answers of organizations according to criteria and sectors. Source: own research.

With the three best evaluated sectors, certain significant differences can be observed. While with electrotechnical sector, the evaluation of particular criteria (except for the "Partnership and Resources (Pa)" criterion) was the highest of all sectors, large quartile ranges signalized a relatively big variability in respondent's answers. Despite the nearness of the compared three sectors, there are, mainly with the "Processes, products, and services (Pr)" and "Leadership (L)" criteria, significant differences observed to the disadvantage of the mechanical sector. So that we can better understand the mentioned differences between particular sectors, in the further analysis we acceded to individual evaluations of the Integrated safety and Digitalization areas.

Figure 4 proves the differences between Integrated safety and Digitalization areas from the viewpoint of respondents' perceptions.

Values from the area of Integrated safety are, on average, higher almost in all criteria. It is Integrated safety, in which, with the "Partnerships and Resources (Pa)" criterion, almost the lowest values are reached in all three sectors (with an exception of a very low evaluation of the "Leadership (L)" criterion in the mechanical sector). During the evaluation of the Digitalization area in the automotive sector, but for the "Leadership (L)" criterion, in all other criteria approximately the same results were reached. In the electrical sector, the "Partnerships and Resources (Pa)" criterion clearly had a bad evaluation, and in the mechanical sector, these were "Processes, products, and services (Pr)" and "Strategy (S)" criteria.

Interpretation: From a deeper subsequent analysis we can observe certain significantly different evaluations of particular criteria within various sectors. While the electrical sector has average values of particular criteria relatively distant from each other, in the automotive sector the evaluation of particular criteria is almost the same, with one exception. While in the electrical and automotive sectors the decrease in the evaluation of a particular criteria for compared areas is relatively slight, and there are more significant differences in the mechanical sector.

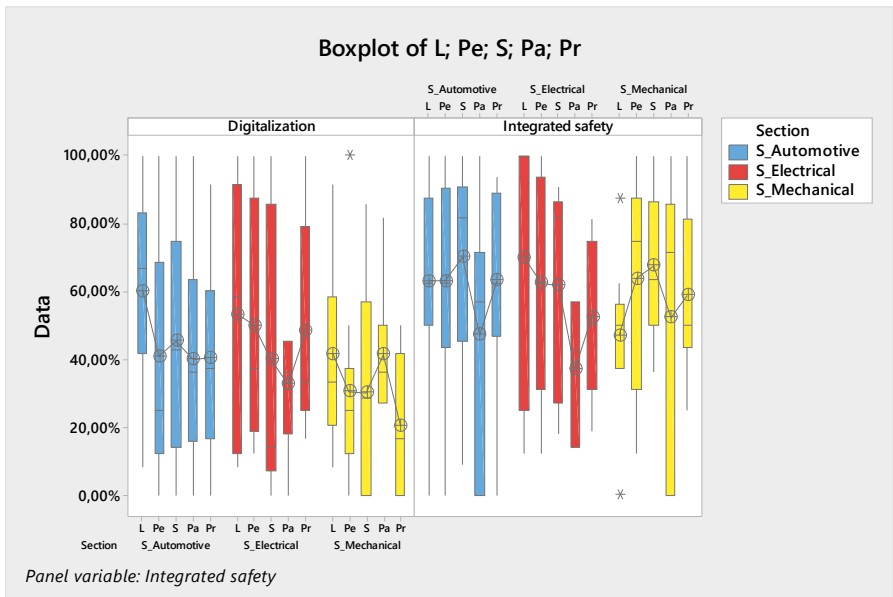

**Figure 4.** Answers of organizations according to area, criterion, and sectors. Source: own research.

Next, a comparison of the remaining sectors was performed, as detailed analysis (segmentation according to areas more or less copies the overall development) will provide comparisons stratified according to criteria and particular sectors (segmentation according to areas did not show any significant changes as for interpretation). Figure 5 presents the four worst numerally evaluated sectors by respondents.

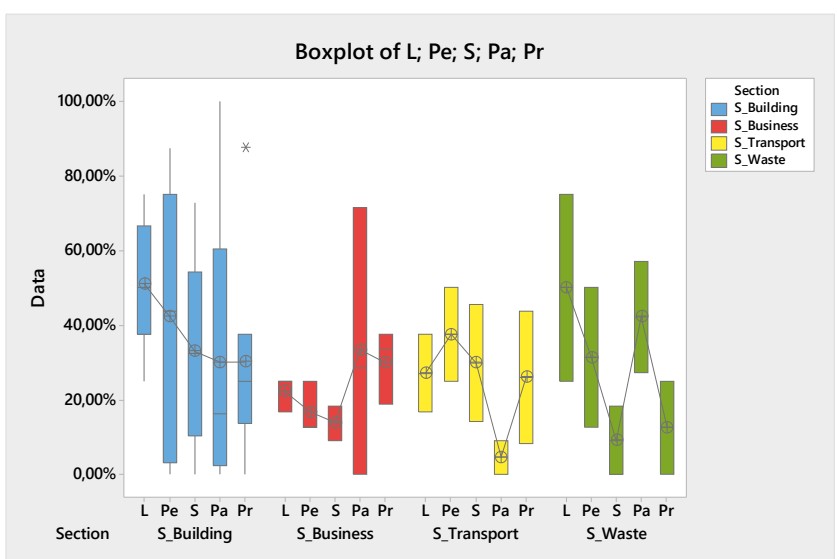

**Figure 5.** Answers of organizations according to criteria and sectors of Building, Business, Transport, and Waste. Source: own research.

Certain distortions may be caused by the relatively small number of respondents in some sectors. In the sector of building, business (commercial—wholesale and retail trade) and waste, the evaluation of the first three criteria has a significantly decreasing trend, and in the case of the business sector it is on a very low level. Evaluation in the transport sector differs by a significantly lower evaluation of the "Leadership (L)" criterion than with "Employees (Pe)" and "Strategy (S)" criteria. "Partnership and Resources (Pa)" is the best evaluated criterion. Paradoxically, a relatively high evaluation is reached with the "Partnerships and Resources (Pa)" criterion in the sector of waste.

Figure 6 presents the results showing sectors which were in the middle of our evaluation. We observe that a certain distortion was caused by relatively small samples of respondents. In the sectors of energy, furniture and services, the "Partnerships and Resources (Pa)" criterion had the worst evaluation. The "Leadership (L)" criterion was, in all sectors, evaluated relatively positively. Low evaluations of the "Processes, products, and services (Pr)" criterion can be observed in the sectors of IT, metallurgy and service.

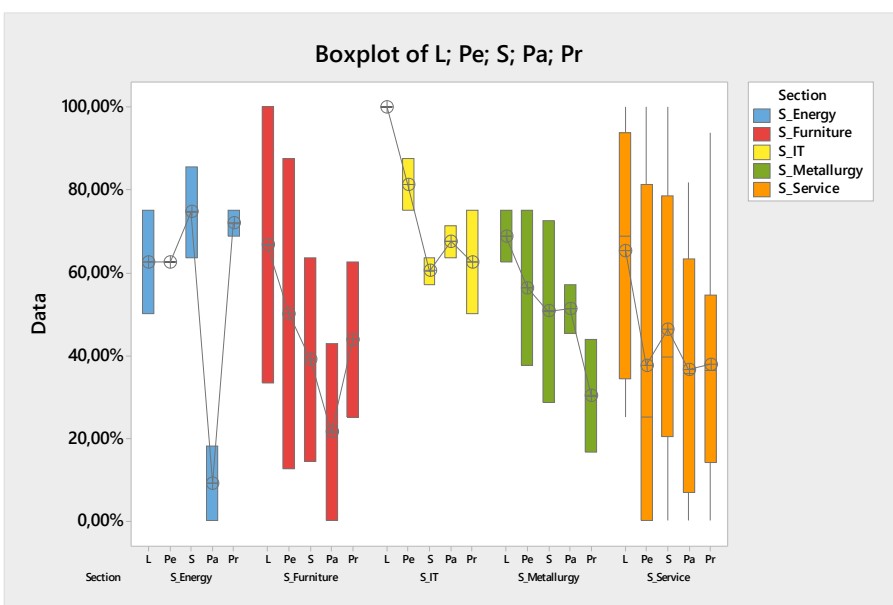

**Figure 6.** Answers of organizations according to criteria and sectors of Energy, Furniture, IT Metallurgy and Service. Source: own research.

Definitely, the best evaluation in this group reaches the IT sector. Although it is a relatively small sample, the perception of Industry 4.0 is evaluated positively, respondents in the IT sector see certain reserves from the "Strategy (S)" criterion. In the energy sector, the "Partnerships and Resources (Pa)" criterion is evaluated very negatively. As the evaluation of other criteria is much higher, we can interpret the result of questioning as respondents' concern from the fact that partners will negatively influence the relatively high readiness of their organizations for the implementation of Industry 4.0. Similar trends, but ones not so strong, can possibly be observed in the furniture sector. It is possible to observe, here, a high variability of particular respondents' answers. It is the same in the case of the service sector. The perception of Industry 4.0 in the IT sector seems highly optimistic and in the metallurgy sector, "Processes, products, and services (Pr)" is the weakest perceived criterion.

## 4. Discussion

The fourth industrial revolution is tightly connected with the degree of automation, which is used in the organization in question. Therefore, the perception of an organization's readiness for Industry 4.0 may be directly influenced by the perception of the automation degree in the researched organization. For this reason, in the next step, we stratified particular respondents' answers into groups according to

how they perceive the ratio between manual and automated production in their organizations. Figure 7 presents Boxplot diagrams of numerical evaluations of particular criteria by respondents divided into groups according to their perception of the automation degree. The ratio *APx:MPy* represents the group of answers by respondents from organizations, which (as these respondents mentioned themselves) have *x*% automated and *y*% mechanical production.

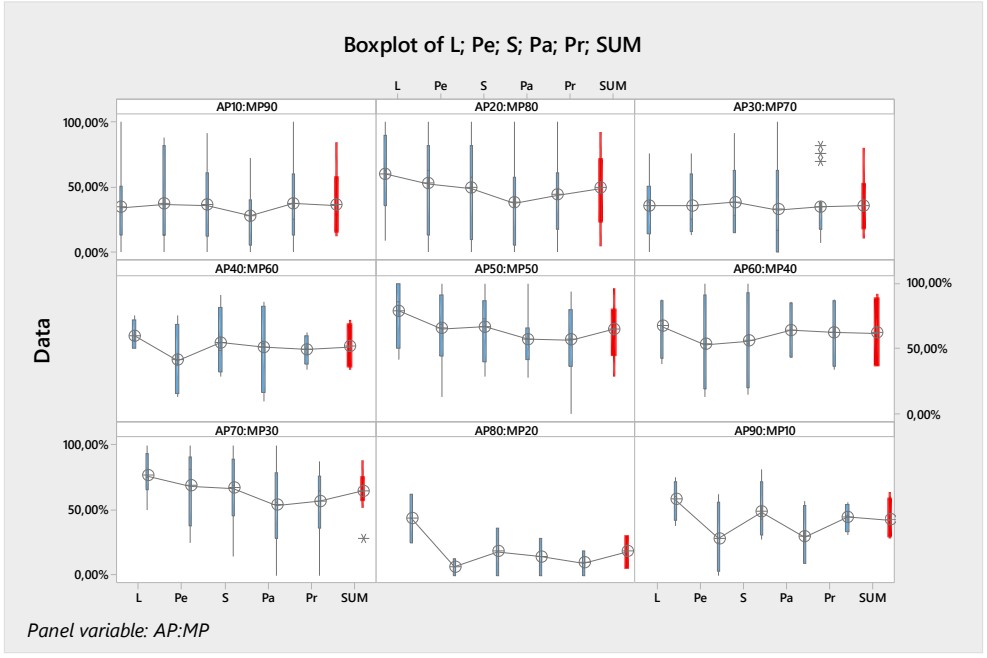

**Figure 7.** Answers of organizations to the ratio of automated and mechanical production. Source: own research.

Paradoxically, the lowest degree of readiness is perceived by respondents in organizations with a high degree of automation. The worst was the evaluation with 80% automation followed by the evaluation with 90% automation. Relatively low evaluations can be observed in all criteria, there are relatively big reserves in organizations with "Employees (Pe)" criterion. Another group with a low evaluation is a group of organizations which have the lowest degree of automation, i.e., 10%. The evaluation of all criteria in this group is approximately on the same level (approximately 40%). In groups with the ratio AP50:MP50 to AP70:MP30, the overall evaluation of readiness is the highest. The evaluation in these three groups has approximately the same behavior, according to which "Leadership (L)" criterion is evaluated the best. All other criteria are evaluated approximately the same, with an average of approximately 60% of the total evaluation which could possibly be reached. In groups with the ratio AP30:MP70 and AP40:MP60, the evaluation was not influenced by big amplitudes of middle values; however, it was possible to a observe generally lower evaluation of readiness.

From a general point of view, the highest readiness for Integrated safety and Digitalization in relation with safety was shown in organizations which had a ratio of automation from 50% to 70%. In all these organizations, the "Employees (Pe)" criterion has the best evaluation.

It is shown, however, that with an increase in automation, the perception of organizational readiness for Industry 4.0 changes in the "Employees (Pe)" criterion, which has the worst evaluation from the viewpoint of their readiness. It may be interpreted as uncertainty coming from the ability of employees to adapt to new requirements in an I4.0 workplace (e.g., cooperation with collaborative robots, simulation-visualization, robot's maintenance, etc.) Furthermore, in organizations with the highest evaluation of the "Employees (Pe)" criterion, this evaluation is lower than the one of the "Leadership (L)" criterion; however, certain concerns are also reported with the evaluation of the

"Partnership and Resources (Pa)" criterion. As it could be expected, the most trust towards "Employees (Pe)" and "Processes, products, and services (Pr)" criteria was shown by organizations with the lowest degree of automation. In contrast to the evaluation by particular sectors, with the division of respondents into groups and according to the automation degree, it is possible to observe in their answers a relatively lower variability of mean values for particular criteria.

Even though Slovakia is a relatively small country, the sample of 53 organizations cannot be considered representative enough, mainly in cases when particular organizations were stratified into various subgroups for the reason of self-assessment. The many sectors in the survey were not statistically sufficiently covered, and, therefore, there were such high dispersions of results. Yet, on the other hand, the results obtained may be understood as the first iteration of further research, which provides a rough estimate of the readiness degree of Slovak organizations for Industry 4.0 from the viewpoint of Integrated safety and Digitalization in relation to occupational health and safety (OHS).

**Author Contributions:** Application of statistical, mathematical techniques, writing and final review of the paper, R.T.; Management and validation, J.S.; Development of methodology, writing and final review, H.P.; Research, data collection, final review of the paper, Z.K. and J.G. All authors have read and agreed to the published version of the manuscript.

**Funding:** This research was funded by: Ministry of Education, Science, Research and Sport of the Slovak Republic APVV No. 15-0351; and Ministry of Education, Science, Research and Sport of the Slovak Republic KEGA No. 015TUKE-4/2019; and Research & Development Operational Program funded by the ERDF ITMS: 26220220182.

**Acknowledgments:** This contribution is the result of the projects implementation: APVV No. 15-0351 Development and application of risk management models in terms of technological systems in line with the industry (Industry) 4.0, and KEGA No. 015TUKE-4/2019 Audit management using software application according to standard ISO 9001:2015 and "University Science Park TECHNICOM for Innovation Application Supported by Knowledge Technology, ITMS: 26220220182, supported by the Research & Development Operational Program funded by the ERDF".

**Conflicts of Interest:** The authors declare no conflict of interest.

## Appendix A

**Table A1.** Questionnaire items based on modified EFQM methodology.

| Criterion | Analysis of Integration of Complex Safety (Safety and Security) Into Management Systems—ISMS | Max Points | Frequency Scale a | b | c | d | e |
|---|---|---|---|---|---|---|---|
| L | Does the top management encourage the idea of integrated safety in your organisation? | 4 | 6 | 3 | 19 | 9 | 16 |
| L | Is a responsible representative for integrated safety management appointed? | 4 | 6 | 9 | 13 | 8 | 17 |
| Pe | Is there a procedure created for training in the area of integrated safety? | 4 | 8 | 7 | 11 | 10 | 17 |
| Pe | Does the requirement for education apply to all levels of organisation management? | 4 | 8 | 5 | 12 | 7 | 21 |
| S | Are there plans for integrated safety implementation? | 4 | 6 | 7 | 9 | 21 | 10 |
| S | Are the implementation plans regularly revaluated and corrected? | 4 | 9 | 11 | 6 | 27 | - |
| S | Are the integrated safety implementation plans in accordance with process management (or particular implementation steps identifiable according to map of processes)? | 3 | 7 | 6 | 8 | 17 | 18 |
| Pa | Have conditions been created for the cooperation with an external organisation in the area of integrated safety implementation? | 4 | 16 | 4 | 14 | 19 | - |
| Pa | Are the activities/areas provided by the external organisation clearly defined? | 3 | 14 | 10 | 16 | 5 | 8 |

**Table A1.** *Cont.*

| Criterion | Analysis of Integration of Complex Safety (Safety and Security) Into Management Systems—ISMS | Max Points | Frequency | | | | |
|---|---|---|---|---|---|---|---|
| | | | Scale | | | | |
| | | | a | b | c | d | e |
| **Pr** | Is there an implemented process of OHS and Security management system integration into management systems? | 4 | 3 | 10 | 13 | 12 | 15 |
| **Pr** | Is there a documented risk management process as a basic tool of management systems (RbT)? | 4 | 5 | 8 | 18 | 8 | 15 |
| **Pr** | Is the process improvement based on integrated approach? | 4 | 2 | 8 | 17 | 8 | 18 |
| **Pr** | Is there a software support oriented to integrated safety? | 4 | 15 | 10 | 8 | 10 | 10 |
| **Analysis of Impact of Digitalization on OHS–OHSd** | | | | | | | |
| **L** | Does the management encourage the implementation of automation and digitalization for OHS support? | 4 | 2 | 8 | 12 | 12 | 19 |
| **L** | Is there a responsible representative for OHSd support appointed? | 4 | 18 | 3 | 9 | 10 | 13 |
| **L** | Are there OHS politics and goals defined, in which automation and digitalization implementation are reflected? | 4 | 16 | 5 | 11 | 14 | 7 |
| **Pe** | Is there employee training provided in the area of OHSd implementation? | 4 | 12 | 17 | 9 | 8 | 7 |
| **Pe** | Does the requirement for education in OHSd apply to all levels of organisation management? | 4 | 19 | 13 | 9 | 4 | 8 |
| **S** | Are there plans for OHSd implementation? | 4 | 17 | 11 | 10 | 11 | 4 |
| **S** | Are the implementation plans regularly revaluated and corrected? | 3 | 19 | 17 | 6 | 11 | - |
| **Pa** | Is the requirement for OHSd communicated and revaluated with involved parties? | 4 | 9 | 12 | 12 | 9 | 11 |
| **Pa** | Have conditions been created for the cooperation with an external organisation in the area of OHSd implementation? | 4 | 28 | 13 | 8 | 4 | - |
| **Pa** | Are the activities/areas provided by the external organisation in OHSd clearly defined? | 3 | 27 | 7 | 10 | 5 | 4 |
| **Pr** | Is there an OHSd implementation process based on process approach? | 4 | 13 | 17 | 12 | 5 | 6 |
| **Pr** | Is the improvement of OHSd processes based on integrated approach? | 4 | 18 | 10 | 18 | 2 | 5 |
| **Pr** | Are there sources for the provision of I4.0 elements into OHSd? | 4 | 19 | 15 | 9 | 8 | 2 |

Note: L—Leadership; Pe—People; S—Strategy; Pa—Partnership and Resources; Pr—Processes, Products and Services; ISMS—Integration of complex Safety (Safety and Security) into Management Systems; OHSd—impact of digitalization on OHS; RbT—Risk based Thinking; OHS—Occupational Health and Safety; a—strongly disagree; b—rather disagree than agree; c—neither agree nor disagree; d—rather agree than disagree; e—strongly agree.

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
