# Peer review of "Application of the EFQM Model to Assess the Readiness and Sustainability of the Implementation of I4.0 in Slovakian Companies"

_sustainability, doi:10.3390/su12145591_

Round 1

Reviewer 1 Report

Table 1. personal or personnel?

line 208: big vs "significant"

line 285: should be "with the degree of automation"

line 319: should be "lowest degree of automation"

line 314: "mistrust that workers will be able to implement? It could be that the potential gains from I4.0 are elusive or not large enough, hence not worth the effort. Suggests a failure on the part of the employees versus management, which might be a poor assumption on part of the authors. Unless the survey asked about mistrust, this is a stretch for a conclusion, suggest rewording, or adding more meaning.

Author Response

All the performed changes were marked in the text by red color.

Reviewer 2 Report

Minor changes and suggestions are written in Comments in the document - I recommend considering them and modify the text. As the results concern - I suppose that many branches (sectors) in the survey were not statistically sufficiently covered and therefore there were such high dispersions of results.

I think this is the first phase and needs more detailed surveys and analysis in future research. However, the results of the survey give an actual picture of readiness for I4.0 in the specified areas (safety and digitalization in OHS) in companies in Slovakia.

Author Response

(The authors gave the same response as above.)
